

# Motor competence of children in Singapore using Movement ABC-2 test

Jernice S.Y. Tan and Michael Chia

Physical Education and Sports Science, National Institute of Education, Nanyang Technological University, Singapore

## ABSTRACT

**Introduction**. The Movement Assessment Battery for Children, 2nd Edition (MABC-2) test is a valuable tool for identifying motor delays in children globally. However, there has been a lack of data for children in Singapore.

**Methods**. This study compared 175 children in Singapore (SG) aged 3–6 years to MABC-2 data based on the United Kingdom (UK) population, using statistical tests to analyze age, country, and gender effects on motor competence.

**Results**. The results showed a positive age effect across all tasks, with SG children excelling in balance but lagging in aiming and catching tasks compared to UK children. The interaction between age and country yielded mixed results, favoring different groups at various ages. Additionally, girls displayed better manual dexterity and static balance than boys.

**Conclusion**. Encouraging more physical activities, especially those involving object manipulation, is crucial for SG children. Establishing local norms for the MABC-2 in Singapore and considering country-specific factors in motor competence evaluations can enhance early intervention strategies. These findings emphasize the importance of tailored approaches to address motor skill development in SG children.

## INTRODUCTION

### Motor competence as the foundation for a healthy and physically active life

Motor competence is vital for children's overall development, impacting cognitive, perceptual, and social aspects, especially in early childhood. The development of motor skills begins in infancy and continues throughout life, with fluctuations in competency levels. Early childhood, up to 6 years old, is a critical period for motor skills development, encompassing effective control and coordination of body movements, including gross and fine motor skills. Research emphasizes the importance of promoting motor skills development in early childhood to establish a strong foundation for future growth and learning, highlighting the significance of tailored approaches to address motor skills development in children (*Bautista et al., 2020*; *Clark & Metcalfe, 2002*; *Hurtado-Almonacid et al., 2024*; *Libertus & Hauf, 2017*).

    Motor competence is fundamental for future physical activity engagement, with research linking it to health and physical fitness (*Chen et al., 2023*; *Stodden et al., 2008*). *Stodden et*

Corresponding author
Michael Chia,
michael.chia@nie.edu.sg

*al. (2008)* proposed a model suggesting that physical activity in early childhood influences motor competence and health-related fitness, which in turn, impacts health-related physical fitness in later childhood. Recent studies have highlighted a bidirectional positive relationship between motor competence and physical fitness in preschoolers, emphasizing the importance of integrating both aspects in early childhood physical education programs (*Chen et al., 2023*). This underscores the significance of developing motor skills early on to establish a foundation for physical activity and overall health throughout childhood and beyond (*Clark & Metcalfe, 2002*; *Hurtado-Almonacid et al., 2024*).

Motor competence assessment tools are crucial for evaluating children's motor skills (*Henderson & Sugden, 1992*; *Henderson, Sugden & Barnett, 2007*). Regular evaluation of motor competence is essential for monitoring children's development and tailoring interventions to enhance their motor skills and overall well-being. These assessments play a vital role in promoting physical activity, health, and successful motor skill development in children.

## Country differences in motor competence in early childhood

Country differences were commonly reported when assessing the motor competence of children of western and eastern descents, including Singapore (*Wright et al., 1994*), Sweden (*Rosblad & Gard, 1998*), Japan (*Miyahara et al., 1998*; *Ruiz et al., 2003*), Hong Kong (*Chow, Henderson & Barnett, 2001*; *Chow et al., 2006*), Spain (*Ruiz et al., 2003*), Taiwan (*Chow et al., 2006*), and Australia (*Livesey, Coleman & Piek, 2007*). Mixed findings were observed in the comparison of Asian children with the US norms of the 1st version of Movement Assessment Battery for Children (MABC). For example, Hong Kong Chinese children of age band 1 (aged 4–6 years) were reported to have significantly lower manual dexterity scores and higher balance scores, indicating better competence in both components, but lower ball skills scores, indicating poorer competence than their US counterparts (*Chow, Henderson & Barnett, 2001*). Later, *Chow et al. (2006)* also observed significant differences in some MABC tasks between Hong Kong and Taiwan preschoolers at age band 1 (aged 4–6 years), which suggested that country differences can exist even within the same ethnic group. In *Miyahara et al.*'s (*1998*) study with older children, Japanese children generally had lower manual dexterity scores than the US norms at age band 2 (aged 7–8 years) and age band 3 (aged 9–10 years), but higher balance scores than the US norms at age band 3 (aged 9–10 years). However, in another study by *Ruiz et al. (2003)* of the same age bands, they did not observe the exact same trend between Japanese children and the US norms. In fact, motor competence differences vary across countries, age bands, MABC motor components, and genders. Additionally, *Ruiz et al. (2003)* suggested that norm adjustments may be needed when using the MABC in different countries, based on the significant interaction effects among Japan, the US, and Spain.

In more recent years, studies using the MABC-2 have also observed country differences such as Brazil (*Valentini, Ramalho & Oliveira, 2014*), China (*Hua et al., 2013*; *Ke et al., 2020*), Croatia (*Serbetar, Loftesnes & Mamen, 2019*), Czech Republic (*Psotta & Hendl, 2012*; *Psotta et al., 2012*), Germany (*Wagner et al., 2011*), Greece (*Ellinoudis et al., 2011*), Israel (*Engel-Yeger, Rosenblum & Josman, 2010*), Italy (*Zoia et al., 2019*), Japan (*Hirata et al.,*

*2018*; *Kita et al., 2016*), Spain (*Navarro-Patón et al., 2021*) and Chile (*Hurtado-Almonacid et al., 2024*). Similar to studies using the MABC, mixed findings were also detected between Asian children and the United Kingdom (UK) norms of the MABC-2. Within age band 1 (aged 3–6 years), Japanese children scored significantly better in manual dexterity and balance skills but significantly poorer in ball skills than UK norms (*Hirata et al., 2018*). A study of older Japanese children of age band 2 (aged 7–10 years) by *Kita et al. (2016)* showed similar results, but they no longer performed worse in ball skills than the UK children and performed significantly better in overall total MABC-2 scores. However, another trend was observed in *Ke et al.*'s study *2020* who reported that Chinese children at age band 1 had better posting coin skills but poorer drawing skills than UK children. In terms of throwing ball on target and one-leg balance, Chinese children performed worse at the younger age of 3 years but improved to perform better at the older ages of 5 and 6 years compared to the UK norms.

Over the past three decades, research on motor competence in children has highlighted country-specific variations (*Chow et al., 2006*; *Engel-Yeger, Rosenblum & Josman, 2010*; *Hua et al., 2013*; *Kita et al., 2016*; *Ruiz et al., 2003*). Recent international studies have consistently shown the impact of age, country, and/or gender on MABC-2 outcomes (*Hirata et al., 2018*; *Hurtado-Almonacid et al., 2024*; *Ke et al., 2020*; *Zoia et al., 2019*). While studies have explored the influence of age, country, and gender on motor competence using tools like the MABC-2, data specific to Singapore and Southeast Asian countries are limited. *Wright et al. (1994)* assessed the MABC checklist's applicability in Singapore for identifying children with movement problems through teacher responses but did not investigate the motor competence of children in Singapore using the actual MABC tasks. In addition, *Hadwin et al. (2023)* stressed the importance of considering tester reliability and environmental factors when interpreting MABC-2 scores. This study aimed to investigate the motor competence of children in Singapore using the MABC-2, comparing our findings to UK data to provide valuable insights for future Southeast Asian studies and inform early interventions to support children's motor development and overall well-being.

## MATERIALS & METHODS

### Research overview

This study was part of a larger project titled "The Development of the Motor Skills of Young Children—Using Traditional Games", which was completed before the COVID-19 pandemic. There were four research phases in total, and all children participated in Phase 1. The data presented in this manuscript were collected and analyzed during Phase 1 and was exploratory in nature at this phase. Each child was also involved in one additional research phase (either Phase 2, 3, or 4) and was randomly assigned to either the intervention group or the control group for that phase.

### Participants

One hundred and seventy-five children (SG children: 90 females, 85 males, aged 3–6 years; see Table 1) schooled in Singapore preschools participated in this study at Phase 1. The demographic characteristics of the participants were suitably representative in terms of

Table 1  Number of SG children in each age and gender group.

| Age (years) | Female | Male | Overall |
|---|---|---|---|
| 3 | 26 | 26 | 52 |
| 4 | 28 | 26 | 54 |
| 5 | 25 | 23 | 48 |
| 6 | 11 | 10 | 21 |
| Total | 90 | 85 | 175 |

geographic region, preschool type, age, and gender. All children reported being healthy with no known medical conditions or disabilities. Ethical approval for the research was obtained from the institutional review board (Name of granting organization: Republic Polytechnic, Singapore; IRB reference code: SHL-M-2014-016). Before data collection began, parental consent and child assent were obtained using an IRB approved informed consent form.

## Instrument & tasks

The Movement Assessment Battery for Children, 2nd Edition (MABC-2; *Henderson, Sugden & Barnett, 2007*) is a well validated assessment tool for children aged 3–16, evaluating both fine and gross motor skills. Its simplicity and short duration make it popular for identifying motor delays and guiding interventions (*Hadwin et al., 2023*). In this study, only the MABC-2 age band 1 (3–6 years old) tasks were used. The MABC-2 tool consists of 8 tasks categorized into 3 components: Manual dexterity (MD1—posting coins, MD2—threading beads, MD3—drawing trails), Aiming & catching (AC1—catching beanbags, AC2—throwing beanbags onto target), and Balance (BAL1—one-leg balance, BAL2—walking heels raised, BAL3—jumping on mats).

## Procedures

The study took place indoors, with a circle time activity conducted to familiarize the children with the testers and their environment. Test sequences were randomized to mitigate novelty and fatigue effects. The participants were tested either with or without footwear. Footwear use was optional with no impact on stability and locomotion tasks (*Tan, 2019*). Trained testers provided standardized instructions and demonstrations. Testers achieved high intra- and inter-tester reliability (intra-class coefficient index > 0.7; *Shrout & Fleiss, 1979*) through familiarization training and reliability videos. Each child's assessment took approximately 20 minutes. These meticulous procedures ensured consistency and accuracy in evaluating motor competence in SG children, contributing valuable data for future research and interventions in Southeast Asia.

## Data reduction & analysis

Each child was scored according to the MABC-2 task protocols. Raw scores (RS) were calculated using the best attempt of each task. Ten RS were recorded, except in cases of failure or refusal. The RS for MD1 and BAL1 were averaged for the preferred/non-preferred hand and the best/other leg respectively to derive 8 RS. For MD tasks, lower scores indicate

higher competence, while for AC and BAL tasks, higher scores indicate higher competence. With MABC-2 authors' permission under the Pearson standarisation data agreement, we compared data from our study to anonymized MABC-2 raw data, based on the UK population. A three-way multivariate analysis of variance (MANOVA) test, an analysis of variance (ANOVA) test for between-subjects factors and *post-hoc* tests were conducted on the three independent variables (country, age and gender) and the eight dependent variables (MD1, MD2, MD3, AC1, AC2, BAL1, BAL2 & BAL3) to evaluate any significant country, age, and gender differences. The effect size values denoted by $\eta_p^2$ for low, moderate, and high were 0.01, 0.06 and 0.14, respectively (*Cohen, 1988*). The level of significance for all statistical analyses was set at $p < .05$.

## RESULTS

Descriptive analysis presents the means and standard deviations of SG children's raw scores of eight tasks of MABC-2 age band 1 (see Table 2). A main age effect was found, with older children generally performing significantly better than the younger ones with high effect size ($F_{(24, 1758)} = 34.531$, $p < .001$, Pillai's Trace $= 0.961$, $\eta_p^2 = .320$; see Table 3). From ages 3 to 4, significant positive age effect was observed for all eight tasks (see Tables 2 and 4). From ages 4 to 5, children performed significantly better in six out of eight tasks — all manual dexterity tasks (MD1, MD2 & MD3), throwing beanbag onto mat (AC2) and most balance tasks (BAL1 & BAL2). From ages 5 to 6, children scored better in only four out of eight tasks — posting coins (MD1), threading beads (MD2), catching beanbag (AC1) and one-leg balance (BAL1). The non-significant age effect exceptions were seen from ages 4 to 5 for AC1 and jumping on mats (BAL3), and from ages 5 to 6 for drawing trail (MD3), AC2, walking heels raised (BAL2) and jumping on mats (BAL3) (see Tables 2 and 4).

A main country effect between SG and UK children was observed on five out of eight tasks with high effect size ($F_{(8, 584)} = 21.797$, $p < .001$, Pillai's Trace $= .230$, $\eta_p^2 = .230$; see Table 3). The SG children scored significantly higher for all balance tasks (BAL1, BAL2 and BAL3) while the UK children exhibited higher scores for all ball tasks (AC1 and AC2; see Tables 2 and 3). A main gender effect was also observed on three out of eight tasks with moderate effect size ($F_{(8, 584)} = 4.690$, $p < .001$, Pillai's Trace $= .060$, $\eta_p^2 = .060$). The girls had lower MD2 and MD3 scores and higher BAL1 scores, indicating better competence for most manual dexterity tasks and balance on one leg (see Tables 2 and 3).

Significant age × country interaction with low effect size ($F_{(24, 1758)} = 2.304$, $p < .001$, Pillai's Trace $= 0.091$, $\eta_p^2 = .030$) was reported but not for age × gender, country × gender and age × country × gender interactions. The SG children did better than their UK counterparts for drawing trail (MD3) at ages 3 and 4 years, but worse at ages 5 and 6 years. For catching beanbag (AC1), the SG children scored consistently poorer at ages 3, 4, 5 and 6 years compared to the UK children. Lastly for balance on one leg (BAL1), the SG children did significantly better at ages 3, 5 and 6 years but not at the age of 4 years (see Tables 2 and 3).
**Table 2  Descriptive statistics of Singaporean children on MABC-2 tasks in Age Band 1.**

| Age Band 1 | 3 years old | | 4 years old | | 5 years old | | 6 years old | | All (gender) | |
|---|---|---|---|---|---|---|---|---|---|---|
| | M | SD | M | SD | M | SD | M | SD | M | SD |
| MD1: posting coins | | | | | | | | | | |
| Female | 13.58 | 2.45 | 12.29 | 2.81 | 20.12 | 2.07 | 19.09 | 1.51 | 15.67 | 4.17 |
| Male | 14.96 | 2.42 | 12.27 | 2.29 | 20.57 | 2.02 | 19.20 | 1.14 | 16.15 | 4.01 |
| All (age) | 14.27 | 2.51 | 12.28 | 2.55 | 20.33 | 2.04 | 19.14 | 1.31 | 15.90 | 4.09 |
| MD2: threading beads | | | | | | | | | | |
| Female | 42.35 | 11.01 | 35.75 | 11.17 | 50.64 | 12.06 | 47.18 | 8.93 | 43.19 | 12.46 |
| Male | 57.19 | 22.71 | 38.35 | 8.65 | 53.35 | 9.72 | 47.40 | 6.98 | 49.24 | 16.34 |
| All (age) | 49.77 | 19.19 | 37.00 | 10.03 | 51.94 | 10.97 | 47.29 | 7.86 | 46.13 | 14.75 |
| MD3: drawing trail | | | | | | | | | | |
| Female | 4.04 | 3.57 | 1.82 | 2.50 | 0.56 | 0.82 | 0.91 | 1.38 | 2.00 | 2.80 |
| Male | 5.08 | 4.28 | 2.12 | 2.27 | 1.30 | 1.40 | 0.80 | 0.79 | 2.65 | 3.22 |
| All (age) | 4.56 | 3.94 | 1.96 | 2.37 | 0.92 | 1.18 | 0.86 | 1.11 | 2.31 | 3.02 |
| AC1: Catching Beanbag | | | | | | | | | | |
| Female | 3.04 | 2.39 | 3.86 | 2.34 | 5.88 | 1.86 | 6.82 | 2.56 | 4.54 | 2.62 |
| Male | 2.46 | 2.40 | 4.27 | 2.41 | 5.57 | 2.43 | 7.00 | 2.40 | 4.39 | 2.82 |
| All (age) | 2.75 | 2.39 | 4.06 | 2.36 | 5.73 | 2.13 | 6.90 | 2.43 | 4.47 | 2.71 |
| AC2: Throwing Beanbag onto Mat | | | | | | | | | | |
| Female | 2.73 | 2.24 | 3.46 | 2.05 | 5.00 | 1.94 | 4.73 | 2.05 | 3.83 | 2.24 |
| Male | 2.65 | 1.81 | 4.08 | 1.74 | 4.57 | 2.11 | 5.30 | 2.31 | 3.92 | 2.11 |
| All (age) | 2.69 | 2.02 | 3.76 | 1.91 | 4.79 | 2.01 | 5.00 | 2.14 | 3.87 | 2.18 |
| BAL1: One-leg Balance | | | | | | | | | | |
| Female | 9.12 | 7.03 | 11.39 | 7.55 | 21.88 | 9.11 | 27.45 | 4.08 | 15.61 | 10.09 |
| Male | 5.88 | 4.07 | 8.65 | 7.68 | 21.74 | 8.41 | 23.20 | 8.46 | 13.06 | 10.17 |
| All (age) | 7.50 | 5.92 | 10.07 | 7.67 | 21.81 | 8.69 | 25.43 | 6.73 | 14.37 | 10.18 |
| BAL2: Walking Heels Raised | | | | | | | | | | |
| Female | 10.92 | 4.53 | 11.29 | 4.88 | 14.28 | 2.64 | 15.00 | 0.00 | 12.47 | 4.21 |
| Male | 10.23 | 4.02 | 12.27 | 3.81 | 14.26 | 1.66 | 15.00 | 0.00 | 12.51 | 3.61 |
| All (age) | 10.58 | 4.26 | 11.76 | 4.39 | 14.27 | 2.20 | 15.00 | 0.00 | 12.49 | 3.92 |
| BAL3: Jumping on Mats | | | | | | | | | | |
| Female | 4.42 | 1.06 | 4.54 | 1.10 | 4.92 | 0.40 | 4.73 | 0.65 | 4.63 | 0.91 |
| Male | 4.54 | 0.99 | 4.77 | 0.65 | 4.87 | 0.63 | 5.00 | 0.00 | 4.75 | 0.74 |
| All (age) | 4.48 | 1.02 | 4.65 | 0.91 | 4.90 | 0.52 | 4.86 | 0.48 | 4.69 | 0.83 |

**Notes.**

MD1, average of preferred MD1 & non-preferred MD1 completion time in secs (6 coins for 3 & 4 yrs; 12 coins for 5 & 6 yrs); MD2, completion time in secs.; MD3, *n*. of errors; AC1, *n*. of correct catches out of 10 (by hands or by trapping ball against body for 3 & 4 yrs; only by hands for 5 & 6 yrs); AC2, *n*. of correct throws out of 10; BAL1, average of best & other leg balance time, up to 30 s; BAL2, *n*. of correct steps up to 15; BAL3, *n*. of correct jumps up to 5 (discontinuous jumps, feet can adjust & feet not landing together allowed for 3 & 4 yrs; continuous jumps, feet cannot adjust & both feet must land together for 5 & 6 yrs).

## DISCUSSION

This study aimed to investigate the motor competence of children in Singapore using the MABC-2. By comparing our findings to the MABC-2 raw data based on the United Kingdom (UK) population, we sought to provide valuable data for future Southeast Asian

**Table 3  Three-way MANOVA with univariate *F*-tests of significance for main & interactions effect.**

| Tasks | $F$ | $P$ | $\eta^2_p$ | $F$ | $p$ | $\eta^2_p$ | $F$ | $p$ | $\eta^2_p$ |
|---|---|---|---|---|---|---|---|---|---|
| Main effect | | | | | | | | | |
| | | Age | | | Country | | | Gender | |
| MD1 | 250.74 | 0.001* | 0.560 | 0.78 | 0.376 | 0.001 | 1.29 | 0.256 | 0.002 |
| MD2 | 33.23 | 0.001* | 0.144 | 0.43 | 0.510 | 0.001 | 17.86 | 0.001* | 0.029 |
| MD3 | 63.29 | 0.001* | 0.243 | 3.29 | 0.070 | 0.006 | 6.34 | 0.012* | 0.011 |
| AC1 | 41.71 | 0.001* | 0.175 | 81.02 | 0.001* | 0.121 | 0.00 | 0.995 | 0.000 |
| AC2 | 39.08 | 0.001* | 0.166 | 20.49 | 0.001* | 0.034 | 1.61 | 0.205 | 0.003 |
| BAL1 | 113.62 | 0.001* | 0.366 | 13.88 | 0.001* | 0.023 | 14.85 | 0.001* | 0.025 |
| BAL2 | 37.27 | 0.001* | 0.159 | 9.53 | 0.002* | 0.016 | 1.06 | 0.303 | 0.002 |
| BAL3 | 10.28 | 0.001* | 0.050 | 10.59 | 0.001* | 0.018 | 0.34 | 0.557 | 0.001 |
| Interaction effect | | | | | | | | | |
| | | Age × Country | | | Age × Gender | | | Country × Gender | |
| MD1 | 0.65 | 0.583 | 0.003 | 1.19 | 0.314 | 0.006 | 0.40 | 0.530 | 0.001 |
| MD2 | 0.91 | 0.438 | 0.005 | 2.78 | 0.040 | 0.014 | 0.55 | 0.457 | 0.001 |
| MD3 | 3.52 | 0.015* | 0.018 | 0.69 | 0.557 | 0.004 | 0.76 | 0.384 | 0.001 |
| AC1 | 5.71 | 0.001* | 0.028 | 0.53 | 0.664 | 0.003 | 0.12 | 0.731 | 0.000 |
| AC2 | 0.48 | 0.700 | 0.002 | 1.02 | 0.385 | 0.005 | 0.16 | 0.691 | 0.000 |
| BAL1 | 4.95 | 0.002* | 0.024 | 0.90 | 0.443 | 0.005 | 0.06 | 0.813 | 0.000 |
| BAL2 | 0.97 | 0.408 | 0.005 | 0.52 | 0.667 | 0.003 | 1.47 | 0.225 | 0.002 |
| BAL3 | 2.24 | 0.083 | 0.011 | 1.09 | 0.355 | 0.005 | 4.72 | 0.030 | 0.008 |

**Notes.**
MANOVA analysis showed significant results with * at $p < .05$.

**Table 4  *Post-hoc* tests for multiple age comparisons.**

| Tasks | 3 *vs* 4 years old | | 4 *vs* 5 years old | | 5 *vs* 6 years old | |
|---|---|---|---|---|---|---|
| | SE | $p$ | SE | $p$ | SE | $p$ |
| MD1 | 0.298 | 0.005* | 0.313 | 0.005* | 0.374 | 0.001* |
| MD2 | 1.603 | 0.005* | 1.681 | 0.005* | 2.010 | 0.006* |
| MD3 | 0.327 | 0.005* | 0.343 | 0.005* | 0.410 | 1.000 |
| AC1 | 0.242 | 0.005* | 0.254 | 1.000 | 0.263 | 0.005* |
| AC2 | 0.212 | 0.005* | 0.222 | 0.005* | 0.266 | 0.108 |
| BAL1 | 0.785 | 0.005* | 0.823 | 0.005* | 0.984 | 0.004* |
| BAL2 | 0.408 | 0.005* | 0.428 | 0.005* | 0.511 | 1.000 |
| BAL3 | 0.099 | 0.005* | 0.104 | 1.000 | 0.124 | 1.000 |

**Notes.**
*Post-hoc* tests showed significant results with * at $p < .05$.

studies and inform early interventions to support children's motor development and overall well-being.

Research using either the MABC (*Henderson & Sugden, 1992*) or MABC-2 (*Henderson, Sugden & Barnett, 2007*) revealed mixed findings for country and gender differences, but more consistent findings for age differences. This discussion focused on studies using the

MABC-2, especially those examining age, country, and gender differences, for a more direct and meaningful comparison of studies conducted in different countries.

## Age effects in SG children

Both SG and UK children experienced a positive effect on motor competence with age on all eight tasks as they got older. This trend aligns with previous studies using the MABC (*Chow, Henderson & Barnett, 2001*; *Chow et al., 2006*; *Livesey, Coleman & Piek, 2007*) and MABC-2 (*Ke et al., 2020*; *Zoia et al., 2019*). *Navarro-Patón et al. (2021)*, which reported a significant relative age effect (RAE) in MABC-2 skills among 4- and 5-year-olds, even with a narrow age gap of only 3 months. Notably, the RAE was more salient within the 4-year-old age group in the present study. *Zoia et al. (2019)* offered a possible exception, where Italian children reached a plateau performance (ceiling effect) on a jumping task by age 5 years. Age effects in young children's motor competence stem from various factors, such as developmental milestones, neurological maturation, physical growth, practice, environmental stimuli, social learning, educational interventions, and health status. However, the significantly higher competence of older children was less pronounced in terms of the number of tasks in the older age group comparison (*e.g.* 5 *vs* 6 years old) in the present study.

## Country differences between SG and UK children

Our study found significant differences between SG and UK children. In terms of country differences, in five out of the eight tasks (AC1, AC2, BAL1, BAL2 & BAL3) significant differences were detected. In general, SG children performed better in balance skills, while UK children performed better in ball skills. There were no significant differences in manual dexterity skills. While we did not detect significant differences in manual dexterity skills between SG and UK children, this was observed in Japanese, Chinese, and Italian children (*Hirata et al., 2018*; *Ke et al., 2020*; *Zoia et al., 2019*) in studies conducted elsewhere. It is important to note that while the results of the two cited studies (*Ke et al., 2020*; *Zoia et al., 2019*), including the present study, were based on raw scores, the study on Japanese children (*Hirata et al., 2018*) was based on standard scores converted with the MABC-2 normative tables. With the exception of mixed findings for the drawing trail task (MD3), the Japanese and Chinese children performed better than the UK children in the posting coins task (MD2). However, Italian children performed worse in all manual dexterity tasks at different age periods between 3 and 6 years old (*Zoia et al., 2019*).

Our results highlighted significantly better balance skills in SG children. However, the reverse was seen for ball skills, with SG children scoring lower on the aiming and catching (AC) tasks. This was also seen in studies by *Hirata et al. (2018)* and *Zoia et al. (2019)*, but not in *Ke et al.*'s (*2020*). The Japanese children displayed significantly poorer competence in AC tasks than the UK children of the same gender and age group (*Hirata et al., 2018*). Italian children also performed worse in catching (AC1) at younger ages of 3 and 4 years but caught up with their UK peers at ages 5 and 6 years. Some similar superior balance skills were also seen in studies on Asian children, such as Japan (*Hirata et al., 2018*) and China (*Ke et al., 2020*). However, Italian children scored lower raw scores in walking heel raised (BAL2) than UK children (*Zoia et al., 2019*). In contrast, SG children achieved higher

scores in all three balance tasks when compared to UK children. Similarly, we found similar observations in 2 of the 3 balance tasks in *Ke et al.*'s (*2020*) study, except for BAL2, where Chinese children attained significantly higher scores compared to the UK dataset.

Research consistently highlights the importance of physical activity for motor competence in early childhood (*Chen et al., 2023*; *Robinson et al., 2015*; *Stodden et al., 2008*). It is almost universal for preschools to include physical activity in their schedules. Nonetheless, a potential limitation was observed in play environment, space, type, and/or equipment in Singaporean preschools. While *Bautista et al. (2020)* reported more adequate space outdoors than indoors, more children were engaged in indoor play despite documented benefits of outdoor activity for moderate-to-vigorous physical activity (*Chen et al., 2020*; *Chia et al., 2022*). Factors like limited accessibility of outdoor locations and undesirable weather (*e.g.*, rain, heat) might contribute to this indoor play preference (*Bautista et al., 2020*; *Chen et al., 2020*). *Bautista et al.*'s (*2020*) impact study also found that the most frequently taught or practised gross motor category was stability (non-locomotion), followed by locomotion, and finally object manipulation (including ball skills). Furthermore, the lack of physical space and portable equipment specifically designed for ball skills development likely hinders motor competence (*Bautista et al., 2020*; *Chen et al., 2020*). This suggests that differences in access to suitable play spaces and opportunities between SG and the UK could contribute to the observed variations in balance and ball skills between children in these countries. However, further research is warranted to affirm this assertion.

Classrooms and playgrounds often serve as commonly designated indoor and outdoor play spaces respectively. While in-school time plays a crucial role in physical development, out-of-school activities also merit consideration. The findings by *Chen et al. (2020)* showed that limited exposure to ball activities of preschoolers might explain the performance of children in Singapore. A parental survey by *Chen et al. (2020)* found that indoor gyms were the most prevalent community resource used for preschoolers' physical activity beyond school hours. Playgrounds during in-school time and indoor gyms during out-of-school time seem to be the preferred settings for physical activity engagement by teachers and parents in Singapore. However, again, the usual physical activities offered in these settings, such as running, climbing, jumping, and hopping, primarily focus on building stability and locomotion skills rather than ball skills development.

The significant interaction effect for age by country provides additional insights. Although there was no country effect between SG and UK children in the drawing trail task (MD3), the significant interaction effect suggests a pattern. The SG children had lower drawing errors at ages 3 and 4 years but had higher drawing errors at ages 5 and 6 years compared to their UK counterparts. Under the Nurturing Early Learners Framework, Singapore (*Ministry of Education [MOE], 2022*), all five learning areas, namely "Aesthetics and Creative Expression", "Discovery of the World", "Health, Safety and Motor Skills Development", "Language and Literacy" and "Numeracy" emphasise activities that develop fine motor skills, including drawing, painting, colouring, and writing. This focus on fine motor skills in SG preschool curricula may encourage a head start for children as young as 3 years old to ensure they have the necessary fine motor competence before

entering formal primary school at the age of 7. Similar trends in drawing competence have been observed in Japanese children (*Hirata et al., 2018*), but not in Chinese children (*Ke et al., 2020*). Coincidentally, both SG and Chinese children performed worse in MD3 than UK children at ages 5 and 6 years.

For the balance on one leg task (BAL1), SG children outperformed UK children at ages 3, 5, and 6 years, but not at age 4 years. Conversely, SG children scored consistently lower than UK children on the catching beanbag task (AC1) across all ages (3–6 years), as previously explained and supported by similar findings in studies with this age group (*Hirata et al., 2018*; *Zoia et al., 2019*). Similar reasoning based on prior research on Singaporean preschools (*Bautista et al., 2020*; *Chen et al., 2020*) might explain the better balance skills and lower ball skills observed in SG children. Interestingly, Chinese children showed a different pattern, performing worse than UK children at age 3 years but surpassing them at ages 3.5, 4, 5, and 6 years (*Ke et al., 2020*). The reasons for these contrasting performances between SG and Chinese children at different age stages require further investigation.

## Gender differences in SG children

Our findings revealed that girls outperformed boys in threading beads (MD2), drawing trails (MD3), and one-leg balance (BAL1), suggesting better manual dexterity and static balance in girls. This aligns with previous studies using the MABC-2, which have also reported gender differences in these areas, although not in ball skills. While these studies showed similar significant gender differences in balance tasks (*Hirata et al., 2018*; *Ke et al., 2020*), they also showed significantly lower competence in girls for AC2 (*Ke et al., 2020*). Unlike other studies that often report better aiming and catching (AC) skills in boys (*Ke et al., 2020*), we did not observe this higher competence in AC tasks in boys. A plausible explanation might be that our study population came from a highly urbanized environment with relatively homogeneous upbringing practices across genders. The SG parents frequently engage their children, regardless of gender, using common resources within the community, such as activities in parks, playgrounds, and indoor gyms (*Chen et al., 2020*).

## Strengths and limitations

A key strength of this study lies in the use of the MABC-2, as it is a well-established and reliable tool for assessing motor competence across various countries (*Ellinoudis et al., 2011*; *Hua et al., 2013*; *Serbetar, Loftesnes & Mamen, 2019*; *Valentini, Ramalho & Oliveira, 2014*). While internal consistency and factorial validity are generally positive (*Hua et al., 2013*), specific tasks like manual dexterity, throwing, and jumping might require further investigation within the context of Singapore (*Ellinoudis et al., 2011*; *Hirata et al., 2018*; *Serbetar, Loftesnes & Mamen, 2019*). This study contributes valuable knowledge to the limited research on motor competence of children in Singapore aged 3–6 years. The findings provide valuable insights for practitioners and policymakers in the region. The data can inform the planning of appropriate early intervention movement programmes, serving as a reference for teachers and parents. Furthermore, by identifying potential motor competence disparities, the findings can guide recommendations for early exposure to object manipulation activities that address these needs.

This study has its limitations. The inclusion criteria excluded children with known medical conditions or disabilities, potentially overlooking high-functioning children with disabilities at risk of motor delays or undiagnosed coordination disorders. Future research should be more inclusive, following the health equity model outlined by *Douglas et al. (2019)* to prioritize access for all children. This could involve data collection from children with disabilities. Additionally, testing conditions could not fully account for emotional readiness. Due to preschool schedules, some children were tested before lunch or after naptime, which can affect cooperation and focus, particularly in younger children. Future studies can strive for more standardized testing times to minimize these influences. The most significant limitation is the lack of a localized MABC-2 for Singaporean children. While the MABC-2 offers established reliability and validity, country-specific norms are crucial for accurate comparisons and identification of children at risk for motor delays and coordination disorders (*Hirata et al., 2018*; *Ke et al., 2020*; *Zoia et al., 2019*). This study highlights the need for further research to develop these norms and enhance the applicability of the findings in the Southeast Asian context, particularly with the recent release of the newer MABC-3 version. By addressing these limitations and pursuing the suggested future directions, researchers can build upon this study's findings to create a more comprehensive understanding of motor competence in Southeast Asian children.

## CONCLUSION

Our study revealed significant disparities in motor competence between SG children and the UK sample in two out of three motor components of the MABC-2 test for ages 3–6 years. We recommend early exposure to object manipulation activities for SG children due to lower competence in manual dexterity and aiming & catching tasks. Localized MABC-2 versions may enhance accurate comparisons, advocating for country-specific norms in international motor assessments. This study serves as a foundational step towards standardizing a local MABC-2 version, emphasizing the need for further research to develop a Singaporean-specific MABC-3 test.

### Funding

Jernice S.Y. Tan received the Internal Funding Scheme from the Republic Polytechnic (RP), Singapore and the project reference code is 14RIGO02 under RP. The funders had no role in study design, data collection and analysis, decision to publish, or preparation of the manuscript.

### Grant Disclosures

The following grant information was disclosed by the authors:
Republic Polytechnic (RP), Singapore and the project reference code is 14RIGO02 under RP.

## Competing Interests

The authors declare there are no competing interests.

## Author Contributions

- Jernice S.Y. Tan conceived and designed the experiments, performed the experiments, analyzed the data, prepared figures and/or tables, authored or reviewed drafts of the article, and approved the final draft.
- Michael Chia analyzed the data, authored or reviewed drafts of the article, and approved the final draft.

## Human Ethics

The following information was supplied relating to ethical approvals (*i.e.*, approving body and any reference numbers):

The Republic Polytechnic, Singapore, approved the study (SHL-M-2014-016).

## Data Availability

Raw data, including gender, test age and raw scores of participants' MABC-2 test results, are available as a Supplemental File.

## Supplemental Information

Supplemental information for this article can be found online at http://dx.doi.org/10.7717/peerj.18446#supplemental-information.

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
