# Peer review of "Motor competence of children in Singapore using Movement ABC-2 test"

_PeerJ, doi:10.7717/peerj.18446_

## Round 0.1 · original submission · Major Revisions

Dear Authors,

Please revise the manuscript considering the reviewers´ comments/sugestions.

Thank you.

Best regards.

Reviewer 1 ·

Basic reporting

The manuscript is clear and the language is adequate for understanding. The literature references are sufficient for the aim of the study.
The article structure is correct, tables are representative of analysed data, however, some suggestions are given:
- In the introduction it would be useful to deepen the theme of developmental coordination disorders.
- Line 33. It would be preferable to write "up to 6 years old", instead of "up to age 6 years".
- Line 114. The aim declared "This study aimed to investigate the motor competence of children in Singapore using the MABC-2, providing valuable data for future comparative studies in Southeast Asia and informing interventions for children with DCD" is not aligned with the aim stated in the results' discussion (line 207) "The aim of this study was to examine the effects of age, country, and gender on the motor competence of children in Singapore (SG children) compared to the MABC-2 raw data based on the United Kingdom (UK) population". It would be useful to make these statements coherent

Experimental design

The experimental design is well defined, but the research procedures described in the informed consent document has been partially reported in the manuscript (i.e. in the informed consent document have been considered a control group and an intervention group, but there is no reference to these groups in the manuscript; there is also a reference to the motor competency checklist, that represents a part of the MABC2, but no data has been reported about it).
The research procedure should be more detailed and extensive to allow the replication of the study, and it must be explained why the checklist data are not reported if they have been acquired and, in general, must be justified everything in the manuscript that does not comply with the procedures described in the informed consent document.

Validity of the findings

From the informed consent document, it emerges the research has been carried out from Oct 2014 to Sep 2016, eight years have passed since the survey was carried out, it would be appropriate to have more recent data, considering the trends of motor involution and the impact of the covid pandemic on the motor skills of children.

As the study examined the effects of age, country, and gender on the motor
competence of children in Singapore (SG children) compared to the MABC-2 raw data based on the United Kingdom (UK) population it would be appropriate to report the demographic information about the two samples in a (or more) comparative table(s)

Additional comments

May I suggest not using as keywords, words already stated in the title (i.e. motor compentence, MABC2, children), but highlighting other factors related to the topic discussed (i.e. DCD, manual dexterity, aiming and catching, balance, standardized motor test).

There are some inconsistencies in the manuscript: the discussions deal with both comparisons between Singapore children with UK and other Countries; the comparative analysis is fine, but it is not consistent with the stated objective (line 114) "This study aimed to investigate the motor competence of children in Singapore using the MABC-2, providing valuable data for future comparative studies in Southeast Asia and informing interventions for children with DCD".

Reviewer 2 ·

Basic reporting

The study requires updating the references and citing some lines.

Experimental design

The objective of the study requires clarity.

The type and design of the study is not detailed.

Validity of the findings

he statistical analysis is adequate, the data is well described, the strengths and weaknesses are well stated and the conclusions are clear.

Additional comments

The author center them study on the test. I suggest focus on the motor competence and to compare children of Singapore and United Kingdom. This is critical point in the manuscript.

Annotated reviews are not available for download in order to protect the identity of reviewers who chose to remain anonymous.

---

## Round 0.2 · accepted · Accept

Dear authors,

Thank you for adressing the suggestions provided by the reviewers.

Best regards.

Reviewer 1 ·

Basic reporting

No comment

Experimental design

No comment

Validity of the findings

No comment

Additional comments

The authors have reviewed the article in accordance with the comments and suggestions provided. The article meets the journal standards now.